# Astrogliosis Releases Pro-Oncogenic Chitinase 3-Like 1 Causing MAPK Signaling in Glioblastoma

**DOI:** 10.3390/cancers11101437

**Published:** 2019-09-26

**Authors:** Julian Wurm, Simon P. Behringer, Vidhya M. Ravi, Kevin Joseph, Nicolas Neidert, Julian P. Maier, Roberto Doria-Medina, Marie Follo, Daniel Delev, Dietmar Pfeifer, Jürgen Beck, Roman Sankowski, Oliver Schnell, Dieter H. Heiland

**Affiliations:** 1Translational NeuroOncology Research Group, Medical Center, University of Freiburg, 79106 Freiburg, Germany; julian-wurm@gmx.de (J.W.); simon.behringer@uniklinik-freiburg.de (S.P.B.); vidhya.ravi@uniklinik-freiburg.de (V.M.R.); kevin.joseph@uniklinik-freiburg.de (K.J.); nicolasneidert@hotmail.de (N.N.); julian.maier@uniklinik-freiburg.de (J.P.M.); oliver.schnell@uniklinik-freiburg.de (O.S.); 2Department of Neurosurgery, Medical Center, University of Freiburg, 79106 Freiburg, Germany; roberto.doria@uniklinik-freiburg.de (R.D.-M.); juergen.beck.nch@uniklinik-freiburg.de (J.B.); 3Faculty of Medicine, Freiburg University, 79106 Freiburg, Germany; marie.follo@uniklinik-freiburg.de (M.F.); dietmar.pfeifer@uniklinik-freiburg.de (D.P.); roman.sankowski@uniklinik-freiburg.de (R.S.); 4Department of Medicine I, Medical Center, University of Freiburg, 79106 Freiburg, Germany; 5Department of Neurosurgery, University of Aachen, 79106 Aachen, Germany; ddelev@ukaachen.de; 6Institute of Neuropathology, Faculty of Medicine, University of Freiburg, 79106 Freiburg, Germany; 7Berta-Ottenstein-Programme for Clinician Scientists, Faculty of Medicine, University of Freiburg, 79106 Freiburg, Germany

**Keywords:** astrogliosis, glioblastoma microenvironment, CHI3L1

## Abstract

Although reactive astrocytes constitute a major component of the cellular environment in glioblastoma, their function and crosstalk to other components of the environment is still poorly understood. Gene expression analysis of purified astrocytes from both the tumor core and non-infiltrated cortex reveals a tumor-related up-regulation of Chitinase 3-like 1 (CHI3L1), a cytokine which is related to inflammation, extracellular tissue remodeling, and fibrosis. Further, we established and validated a co-culture model to investigate the impact of reactive astrocytes within the tumor microenvironment. Here we show that reactive astrocytes promote a subtype-shift of glioblastoma towards the mesenchymal phenotype, driving mitogen-activated protein kinases (MAPK) signaling as well as increased proliferation and migration. In addition, we demonstrate that MAPK signaling is directly caused by a CHI3L1-IL13RA2 co-binding, which leads to increased downstream MAPK and AKT signaling. This novel microenvironmental crosstalk highlights the crucial role of non-neoplastic cells in malignant brain tumors and opens up new perspectives for targeted therapies in glioblastoma.

## 1. Introduction

Glioblastoma multiforme (GBM) is the most common primary malignant brain tumor in adults. Due to highly infiltrative growth and invariable recurrence within a short time frame, malignant gliomas are diseases of the entire brain. Investigations on the impact of the tumor microenvironment on malignancy is challenging due to the diverse crosstalk between cell types within the complex central nervous system (CNS) architecture in both health and disease. Astrocytes constitute an invaluable element of the CNS, regulating of a variety of essential processes such as hemostasis, neuronal maturation, and synapse formation [1,2]. In order to respond to pathological alterations of the CNS, astrocytes are able to shift towards reactive states linked to distinct requirements [3]. So far, multiple reactive states of astrocytes have been reported, mainly associated with neurodegenerative and inflammatory diseases or brain trauma [3,4,5,6,7]. Most notably, astrocytes participate in scar formation post brain injury and protect neurons and synapses by the release of neurotrophic factors and thrombospondins [3,5]. In the context of malignant glioma, astrogliosis at the tumor margin was found to express a distinct reactive pattern which aids in the maintenance of an immunosuppressive microenvironment [6]. This reactive state was marked by high expression of CD274 and the glycoprotein Chitinase 3-like 1 (CHI3L1, also termed YLK-40) which has been reported to be released by STAT3 activated astrocytes. Additionally, high levels of pathological CHI3L1 release have been observed in Alexander disease, a leukodystrophy which results in a dysregulation of myelination, caused by a glial fibrillary acidic protein (GFAP) mutation in astrocytes. In this disease, CHI3L1 leads to reduced proliferation in oligodendrocyte progenitor cells and a decrease of myelination [8]. Aside from the numerous degenerative and inflammatory diseases of the CNS [9,10,11], CHI3L1 also plays a decisive role as an oncogenic driver in malignant gliomas [12,13]. In particular, high levels of CHI3L1 in the serum are associated with increased aggressive growth, increased migration, poor response to radio- and chemotherapy, as well as poor survival [12,14,15]. Here, we report that CHI3L1 release is driven by tumor-associated reactive transformed astrocytes. Astrocytic co-culture as well as CHI3L1 treatment causes a transcriptional shift of glioblastoma cells towards a mesenchymal phenotype followed by increased proliferation and migration. Further, CHI3L1 was found to activate the MAPK pathway through IL13Ra2-CHI3L1 binding, with downstream inhibition of the MAPK pathway rescuing the oncogenic capability of CHI3L1.

## 2. Results

### 2.1. Tumor-Astrocyte Marked by Chitinase 3-Like 1 Expression

We start our investigation by the identification of differentially expressed genes between tumor-associated astrocytes versus astrocytes from non-neoplastic specimens, using our previously published dataset (GSE128536). The RNA-seq dataset contains gene expression of astrocytes from both non-neoplastic samples (n = 5) and tumor-associated astrocytes (n = 7). Transcriptional profiles were acquired from astrocytes purified using HepaCAM (hepatic and glial cell adhesion molecule) as a surface marker [6,16] (Figure 1a). We found Chitinase 3-like 1 (CHI3L1) to be the top up-regulated gene in tumor-associated astrocytes (Figure 1b). We then validated our findings by means of immunofluorescence imaging and western blot of HepaCAM-based purified astrocytes from 2 non-malignant cortical and 2 glioblastoma specimens (Figure 1c,d). Due to the limited number of samples in our previous data set, we purified astrocytes from an independent cohort (ctr n = 8, TAA n = 9) and validated the CHI3L1 expression by RT-qPCR, which confirmed a significant increase of CHI3L1 expression in tumor-associated astrocytes. We then analyzed the single-cell RNA-seq dataset released by Darmanis et al., (GSE84465) to validate the heterogeneity and stability of CHI3L1 expression in both glioblastoma cells and astrocytes. Since CHI3L1 is known to be overexpressed in glioblastomas [17], we wanted to assess whether CHI3L1 was globally expressed in glioblastoma tumor cells. Unexpectedly, our results showed that the CHI3L1 expression is mainly linked to a specific cluster of tumor cells and strongly varies between individual patients (Appendix A). In contrast to tumor cells, tumor-associated astrocytes revealed a stable expression of CHI3L1 across all patients (Appendix A). The external data set confirmed the increase of CHI3L1 expression in reactive astrocytes within the glioblastoma environment, Appendix A.

### 2.2. Tumor-Astrocyte Co-Culture Model Confirmed CHI3L1 Release

In order to transfer our findings to a cell culture model, we used a co-culture of primary glioblastoma cell lines and astrocytes tagged either by *Zsgreen* or *mCherry* (Figure 2a). Both naive and co-cultured astrocytes were purified using cell sorting and the transcriptome was analyzed. Our primary goal was to validate if our cell culture model showed similar transcriptional changes, compared to our findings from purified astrocytes. By pairwise comparison, we were able to verify that a large proportion of the genes that were up-regulated in-vivo were also up-regulated in our cell culture model Appendix A. In our latest study, we showed that tumor-associated astrocytes (TAA) are marked by signature genes of alternative activation also referred as the “A2 phenotype” of astrocytes. We used RT-qPCR of signature genes to characterize the transcriptional transformation of co-cultured astrocytes which confirmed a subtype switch towards the alternative activation Appendix A.

We further profiled the secretome of the medium from both naive and co-cultured astrocytes. In order to trace back each cytokine to their originating cell type, we aligned cytokine levels to the expression of either tumor cells or astrocytes. Fluorescence-guided cell sorting was used to separately analyze the transcriptome of isolated astrocytes/tumor cells from naive/co-cultured conditions **(Appendix A**). We identified cytokines that were exclusively upregulated in co-cultured conditions and expressed either in astrocytes or tumor cells (Appendix A). In line with our findings from primary GBM samples, our co-culture model confirmed an increased release of CHI3L1 in co-culture. Further, we measured the time-dependent release of CHI3L1 using an ELISA assay, where we demonstrated that CHI3L1 was released by astrocytes in a time-dependent manner and reached stable levels after 24 h, (Figure 2b,c).

### 2.3. Astrocytic Co-Culture Drive MAPK Pathway Activity and Proliferation in Glioblastoma

In our preceding investigations, we demonstrated that the reactive transformation of astrocytes in our co-culture cell model closely mimics in-vivo circumstances, thereby allowing us to further map alterations in tumor cells caused by astrocytic crosstalk. By extracting gene signatures from all known transcriptional subtypes of glioblastoma (mesenchymal (Mes), proneural (PN), and classical (CL) tumors) as previous published [17], we were able to detect a subtype-switch of the BTSC#168 (classification in recent publication PN/CL [18,19]) cell line towards the mesenchymal phenotype. However, our results also showed that genes of the classical signature were partially up-regulated through astrocytic co-culture (Figure 2d). We then used gene expression data of tumor cells from either naive or co-culture conditions to identify up-regulated pathway activity of co-cultured tumor cells. Gene set enrichment analysis (GSEA) (Figure 2e) showed us a significant enrichment of genes related to the mitogen-activated protein kinase (MAPK) pathway (Figure 2f). It has been previously reported that the MAPK pathway maintains cell survival and promote proliferation [20]. Therefore, we mapped proliferation of co-cultivated tumor cells by the expression of MKI67, a proliferation marker, as well as cell confluence of both naive and co-cultured tumor cells in a time dependent manner. Both findings confirmed that there was a significant increase in proliferation in co-cultured tumor cells (Figure 2g–i). A hallmark of glioblastoma malignancy is its invasive growth pattern. In order to measure tumor cell motion/migration, we used live imaging to track the path of individual cells in both naive and co-culture conditions. Tumor cells in co-culture showed a significant increase in the motion pattern in comparison to naive cultured tumor cells (Figure 2j).

### 2.4. CHI3L1 Treatment Causes AKT and MAPK Activity and Mesenchymal Gene Expression

So far, our results suggest that reactive transformation of astrocytes in the tumor microenvironment result in increased tumor malignancy. We then aimed to investigate the extent to which CHI3L1 release from TAA is linked to effects that were observed in co-cultured tumor cells. To further investigate the mechanism of CHI3L1 interaction, we directly treated a previously characterized proneural (BTSC#168) and mesenchymal (BTSC#2) cell line with 6 µg/mL CHI3L1 (Figure 3a). We first mapped gene expression differences caused by CHI3L1 treatment using RNA-seq. Gene expression analysis showed an up-regulation of genes related to hypoxia, epithelial-to-mesenchymal transition, as well as the p53 pathway, all of which are linked to MAPK signaling [20]. In addition, proneural cells treated with CHI3L1 (BTSC#168) showed a subtype shift towards the mesenchymal phenotype whereas mesenchymal cells (BTSC#2) were not strongly affected by CHI3L1 treatment (Figure 3d). We also profiled the proteome alterations caused by CHI3L1 treatment, resulting in a significant up-regulation of AKT, c-Jun, and ERK phosphorylation, which was confirmed in using western blot analysis (Figure 3e,f). In line with previously published reports, we found an activation of both the AKT and MAPK pathway, which resulted in a subtype shift towards mesenchymal gene expression, mediated by CHI3L1. This identified subtype change is particularly true for proneural tumors but less so for mesenchymal differentiated tumors.

### 2.5. Interleukin 13Ra2 Is the Binding Partner of CHI3L1

Potential binding partners of CHI3L1 have been previously identified, but clear evidence regarding the presence of a receptor present in glioblastoma cells has not been presented (Figure 3g). Literature review helped us identify potential binding partners (CRTH2, TMEM219, and IL13Rα2) and their expression in both non-neoplastic and glioblastoma tissue was analyzed based on data from The Cancer Genome Atlas Program (TCGA) database (Figure 3h). CRTH2 was not strongly expressed in the central nervous system compared to TMEM219 and IL13Rα2. The strongest difference between normal and glioblastoma tissue was shown by IL13Rα2 (Figure 3h), which led to the hypothesis that IL13Rα2 is a particularly tumor-specific binding partner. We performed a co-immunoprecipitation study, which confirmed the binding of CHI3L1 and IL13Rα2 (Figure 3i,j). This finding is in line with a previous report, which identified a CHI3L1-IL13Rα2 binding in oligodendrocyte progenitor cells (OPCs) causing demyelination in the context of Alexander disease. We then measured the proliferation over time in the IncuCyte© system which confirmed increased proliferation due to treatment with CHI3L1 (Figure 4a,b). We further used an JNK inhibitor to validate the downstream inhibition of the MAPK pathway was rescued by the presence of CHI3L1. JNK inhibition led to generally decreased proliferation, suggesting that the MAPK pathway is strongly responsible for proliferation. Additionally, we found that in JNK-inhibited cells, CHI3L1 had no effect on proliferation (Figure 4a,b).

## 3. Discussion

The cellular microenvironment of glioblastoma contains several different cell types such as astrocytes, oligodendrocytes, neurons, and myeloid cells. In the recent past, tumor in general and especially glioblastoma were found to highly interact with its microenvironment [21,22,23,24,25]. The microenvironment was shown to be responsible for multiple malignant properties such as tumor heterogeneity [26,27,28], immune escape properties [21], and metabolic adaptation [18]. The current study focused on the crosstalk between astrocytes and tumor cells within their microenvironment. Cytokines, which were secreted by either astrocytes or tumor cells, allow communication within their microenvironment. The importance of microenvironmental communication via cytokine signaling and its link to tumor malignancy was shown in several recent studies [29,30,31]. Previously, we identified and characterized tumor-associated astrocytes, marked by increased JAK/STAT activation [6], that aids in decreasing inflammation within the tumor environment. The supportive effect of these astrocytes in the promotion of tumor progression and survival is still unclear, and we hypothesize that these reactive astrocytes also play an important role in maintaining malignant properties of the tumor. In this work, we identified a strong release of CHI3L1 caused by tumor-astrocytic crosstalk in tumor specimens as well as in cell culture models. These results are in line with a recent report, suggesting that astrocytes with a GFAP mutation mainly observed in Alexander disease, strongly increased their CHI3L1 release, which affects oligodendrocyte progenitor cells (OPCs) resulting in a loss of myelination and neurological impairment. In general, cell culture models are limited in their meaningfulness due to the artificial environment and the absence of the cellular environment. However, since our results of the cell culture model and primary isolated astrocytes coincide, our cell model can be used to properly simulate the interaction between tumor cells and astrocytes. We showed an increased MAPK and AKT activation, which was observed in both, co-cultured conditions and CHI3L1 treatment of tumor cells. Our results fit with recent reports, showing that MAPK and AKT pathway activation caused by CHI3L1 promote recovery from oxidant injury and inflammasome activation in melanoma metastasis [11]. We also present that CHI3L1 related signal was mediated by CHI3L1-IL13Rα2 binding in tumor cells. Although, the CHI3L1-IL13Rα2 signal was described in OPC, macrophages and metastasis, recent investigations showed that TMEM219 is also involved in the CHI3L1-IL13Rα2 complex to affect MAPK signaling. From the TCGA database, we confirmed that TMEM is also highly present in glioblastoma tissue. While CHI3L1 significantly impacts the survival of glioblastoma patients, TMEM219 did not affect overall survival (Appendix A). Therefore, we hypothesize that CHI3L1 release is mainly responsible for astrocytic promotion of tumor malignancy. Since IL13Rα2 inhibition as a target in glioma therapy proved ineffective, further investigation of the IL13Rα2 signaling and its role in glioma progression needs to be investigated. Newman and colleagues reported a co-activation of proliferation caused by interaction IL13Rα2 and EGFRvIII, a glioblastoma exclusive variant of the EGF receptor. In consideration of the close connection between EGF signal and CHI3L1 mediated signals, astrocytic CHI3L1 release could potentially be co-responsible for a reduced response to EGFR directed therapies. This potential mechanism of resistance needs to be investigated in the future.

## 4. Materials and Methods

### 4.1. Ethical Approval

The local ethics committee of the University of Freiburg approved data evaluation, imaging procedures and experimental design (protocol 100020/09 and 5565/15). The methods were carried out in accordance with the approved guidelines, with written informed consent obtained from all subjects. The studies were approved by an institutional review board.

### 4.2. Cell Culture and Co-Culture Model

Astrocytes (CRL-8621), tumor cell lines (brain tumor stem-like cells [BTSC], BTSC#2, BTSC#168 (provided by Dr. Carro’s lab), purified from surgical specimens) were cultured in serum-free conditions as described previously [18]. For combining different cell types within a co-culture, we used fluorescence tagged cells and culture them under serum-free condition along a time period of 48 h. Cells were plated on laminin-coated dishes as described previously [18]. Validation of co-culture was done after 24 h and 48 h by fluorescence microscopy.

### 4.3. Viral Transduction by Constitutive Reporter Lentiviral Vectors

For whole-cell tracking, tumor and astrocytes, primary cultured glioblastoma cells and astrocytes were transduced with lentiviral particles (astrocytes: pmCherry Vectror, Clonetech, BTSC: pZsGreen1-1 Vector, Clonetech (Mountain View, CA, USA). For the transduction 3 × 10^6^ cells were seed per well and incubated overnight in a 37 °C, 5% CO_2_ incubator. Particles quantity determination was calculated according to the manufacturer’s instructions. The transduction mix was prepared by adding the required volume of thawed viral particles and Polybrene^®^ (800 μg/mL). Medium was changed after 1 day. Quality of transduction was measured after 2 days.

### 4.4. Cell Purification by Immunopanning

Immunopanning was performed according to a modified version of the purification protocol from Zhang et al. [16] The generation of a single-cell suspension from tumor tissue for subsequent immunopanning was achieved by using the Worthington Papain Dissociation System (Worthington Biochemical Corporation, Lakewood, NJ, USA) according to the manufacturer’s instructions. The panning plates were prepared the day before and incubated at 4 °C overnight and washed three times with PBS before adding the cell type specific primary antibody (anti-HepaCAM). The cell suspension was transferred to the HepaCAM plate for incubation. The HepaCAM plate with bound astrocytes were washed eight times with PBS immediately after incubation to remove unspecific bindings. The bound cells were then scraped directly off the panning dish with 1 mL of Qiazol reagent (Qiagen, Venlo, Netherlands). Subsequently, RNA extraction was performed.

### 4.5. Flow Cytometry and Cell Sorting

Astrocytes (mCherry) and tumor cells (ZsGreen) were sorted by FACS Aria III (BD Bioscience, Franklin Lakes, NJ, USA) in the core facility, Medical-Center Freiburg, University of Freiburg, in accordance to the manufacturer’s instructions.

### 4.6. Secretome Profiling

For cytokine analysis, we used the Proteome Profiler Human XL Cytokine Array Kit (R&D Systems, Minneapolis, MN, USA). Cells were cultivated for 48 h in the set-up of either naive- or co-cultured conditions. Medium was harvested and centrifuged to remove particulates. The supernatant was analyzed according to the manufacturer’s protocol. The resulting membranes were developed using ChemiDOC XRS with exposure of 20 min. For the evaluation of the signal intensity, we measured the pixel density with additional software in Image J in accordance to the protein array analyzer macro 36 and post-processed R-software (https://www.r-project.org).

### 4.7. Proteome Profiling

For detecting the relative levels of human protein kinase phosphorylation, we used the Proteom Profiler Human Phospho-Kinase Array Kit (R&D Systems, Minneapolis, MN, USA). Two different cell-lines were cultivated with either recombinant CHI3L1-Protein (6 µg/mL) or without treatment for 48 h. According to the manufacturer’s instructions the cell lysis and the incubation of the nitrocellulose-membranes were performed and subsequently developed using ChemiDOC XRS. The pixel density was measrued by additional software in Image J (https://imagej.nih.gov/ij/) for evaluating the signal intensity. Post-processing by R-Software using AutoPipe-algorithm (https://github.com/heilandd/AutoPipe).

### 4.8. Immunohistochemistry

Cells were plated on laminin-coated cover-slips and cultivated for 48 h and subsequently fixed with 4% paraformaldehyde for 20 min at room temperature. Furthermore, cells were incubated with 20% methanol for 15 min and permeabilized with 0.1% Triton. Blocking was performed with 20% bovine serum albumin (BSA). Binding of the primary antibody at 37 °C for 1 h followed by incubation with secondary antibody for 1 h at room temperature using an appropriate concentration of antibody according to the manufacturer’s datasheets.

### 4.9. Quantitative Real-Time PCR

Primers were produced by life technology (www.lifetechnologies.com). The qRT-PCR reaction was performed using the SYBR Green PCR Master Kit. The PCR reaction was run using the 7900HT Fast Real-Time PCR System with the standard SYBR green protocol. Average cDNA quantities relative to a standard amplified gene (Housekeeper Gene: 18S) were calculated using R-statistics. List of primer is shown in Table 1.

### 4.10. Immunoblotting

Samples were lysed using Radio Immuno Precipitation Buffer (RIPA buffer) and protease inhibitor on ice. Afterwards, the lysate was centrifuged at 14,000 rpm for 30 min at 4 °C. The supernatant was used to measure the concentration by NanoDrop. Laemmli buffer was added to the samples and the concentration was adjusted. For western blotting 4–20% precast gels from BioRad were used. Gels were run for 1.5 h at 92 V. The proteins were transferred from the gel to nitrocellulose membrane. To prevent unspecific antibody binding, the membrane was incubated with blocking buffer for 1 h. The specific antibody was dissolved in 5% BSA TBS-0.1%T buffer, added to the membrane and incubated under constant agitation at 4 °C overnight. The membrane was washed with TBS-0.1%T three times for 10 min and subsequently incubated for 1 h under constant agitation with secondary antibody dissolved in TBS-0.1%T. The membrane was then washed three times for 10 min with TBS-0.1%T and subsequently incubated with BioRad Clarity ECL detecting solution. A digital imager ChemiDoc XRS detected the chemiluminescence emanation from the membrane by transforming the signal into a digital image. Original western blots and additional data were shown in Appendix A.

### 4.11. Co-Immunoprecipitation

Adherent cells were treated with recombinant CHI3L1 for 2 h. Cells were washed with PBS, harvested and resuspended in PBS. The sample was centrifuged at 400× *g* for 5 min. Cells were lysed in lysis buffer (150 mM NaCl, 5 mM Tris pH 7.5, 10% Glycerol, 0.2% NP40, 1 mM EDTA, 1× PMSF, anti-protease, anti-phophatase) for 45 min. The sample was centrifuged at 16,000× *g* for 30 min. Supernatant was separated. Primary antibodies (CHI3L1, Il13Ra2 or IgG) were added, and the samples were incubated overnight at 4 °C under constant agitation. Each sample was processed by using PierceTM Protein A/G Magnetic Beads along to the manufacturer’s instruction. The beads were separated by using a magnetic rack. Subsequently an immunoblot was performed.

### 4.12. Proliferation Measurement by IncuCyte©

Cells were seeded on a 48-well plate (Thermo Fisher Scientific Inc., Waltham, MA, USA) with a density of 4000 cells/well. After an incubation period of 8 h at 37 °C, 5% CO_2_, cells were treated with either CHI3L1 (6 µM), JNK-inhibitor (SP600125, 30 µM), combination of both or left untreated. The IncuCyte^®^ S3 Live-Cell Analysis System (Essen BioScience Inc., Ann Arbor, MI, USA) was used to record fluorescence images hourly for a total time period of 80 h. Optical modules were set up for two channels, green (acquisition time 100 ms) and red (acquisition time 150 ms), using the 10× objective. The obtained data were processed with the device-specific software (Incucyte 2019B) to precisely determine individual cell counts.

### 4.13. RNA Sequencing

The purification of mRNA from total RNA samples was achieved using the Dynabeads mRNA Purification Kit (Thermo Fisher Scientific, Carlsbad, CA, USA). The subsequent reverse transcription reaction was performed using SuperScript IV reverse transcriptase (Thermo Fisher Scientific). For preparation of RNA sequencing, the Low Input by PCR Barcoding Kit and the cDNA-PCR Sequencing Kit (Oxford Nanopore Technologies, Oxford, UK) were used as recommended by the manufacturer. RNA sequencing was performed using the MinION Sequencing Device, the SpotON Flow Cell and MinKNOW software (Oxford Nanopore Technologies) according to the manufacturer’s instructions. Base calling was performed by Albacore implemented in the nanopore software (Oxford Nanopore Technologies). Only D2-reads with a quality score above 8 were used for further alignment. Reads were re-arranged in accordance to their barcode and trimmed by Porechop (https://github.com/-rrwick/Porechop). Alignment was performed by minimap2 (https://github.com-/lh3/minimap2) and processed by sam-tools. Mapped reads were normalized by DESeq2 [32]. The expression matrix was analyzed with AutoPipe (https://github.com/heilandd/AutoPipe) by a supervised machine-learning algorithm and visualized in a heatmap. (Data available in GEO: in preparation).

### 4.14. Microarray Gene Expression

RNA was extracted by All Prep Kit (Qiagen, Venlo, Netherlands) according to the manufacturer’s instructions. RNA integrity was measured using the Agilent RNA Nano Assay Agilent Bioanalyzer 2100 (http://www.home.agilent.com) according to the manufacturer’s instructions. Transcriptome analysis was performed by Affymatrix Clariom D according to the manufacturer’s instructions in the Department of Hematology, Oncology and Stem Cell Transplantation, Medical Center University of Freiburg. (Data available in GEO: in preparation).

### 4.15. Expression Data Analysis

Analysis of differentially expressed genes was performed using the DESeq2 package [32] for RNA-sequencing and limma package for microarray data. DESeq2 mainly uses a generalized linear model with a negative binomial distribution. A detailed description is given in the R documentation. Differentially expression was visualized in volcano plots (*x*-axis: fold-change of expression, *y*-axis: $-log_{10}\left(FDR_{p-value}\right)$) or heatmaps using the AutoPipe package with color code: “Spectral” for proteomic data and “RdBu” for expression data. Further functional analysis was performed by a permutation-based pre-ranked gene set enrichment analysis (GSEA) [33]. The predefined gene sets of the Molecular Signature Database [34] were taken. For significant enrichment, *p*-values were adjusted by FDR. We used the “AutoPipe_tSNE” function for dimensional reduction after clustering (PAM) and cluster optimization by outlier removal. Subtype-switch was determined by ranked gene-expression heatmaps as well as changes of the enrichment score examined by GSEA. For further validation, the enrichment score was projected into a virtual circle in which 1/3 of each circle represents a subgroup. Two trajectories illustrate the affiliation of each sample to a distinct signature subgroup. If a sample is distant from the center, the greater the difference of the enrichment score in the direction of a specific signature, the closer to the neighboring signature areas marks possible overlaps of the scores of different groups. Detailed information of this visualization type is given in our recent publication. In order to analyze associations between gene expression and survival based on the TCGA cohort we used the gliovis tool [35].

## 5. Conclusions

In summary, we investigated a novel environmental mediated mechanism to support tumor progression and malignancy. Reactive astrocytes of the tumor environment released CHI3L1, which drives MAPK and AKT signaling in glioblastoma via stimulation of IL13Rα2. These findings open new perspectives for further investigations and therapeutic approaches in glioblastoma therapy.

## Figures and Tables

**Figure 1 cancers-11-01437-f001:**
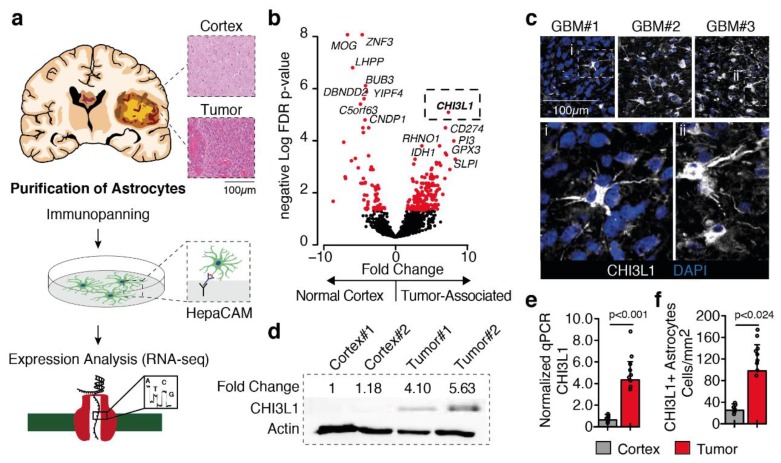
(**a**) Illustration of the workflow. (**a**) Astrocytes from both glioblastoma lesions and non-neoplastic astrocytes were purified. (**b**) Differential gene expression non-neoplastic vs. glioblastoma associated astrocytes. Each dot marks a gene, red dots represent significant differently expressed genes. CHI3L1 was identified as top-up-regulated gene in tumor-associated astrocytes. (**c**,**d**) Immunostaining and western blot of CHI3L1 protein-level confirm our gene expression findings. (**e**) PCR validation of CHI3L1 expression as well as quantification of numbers of CHI3L1 cells (**f**). *p*-values are determined by negative binomial distribution (DESeq) (**b**) Wilcoxon sum rank test (**e**,**f**) adjusted by Benjamini–Hochberger for multiple testing. Data is presented as mean ± standard deviation.

**Figure 2 cancers-11-01437-f002:**
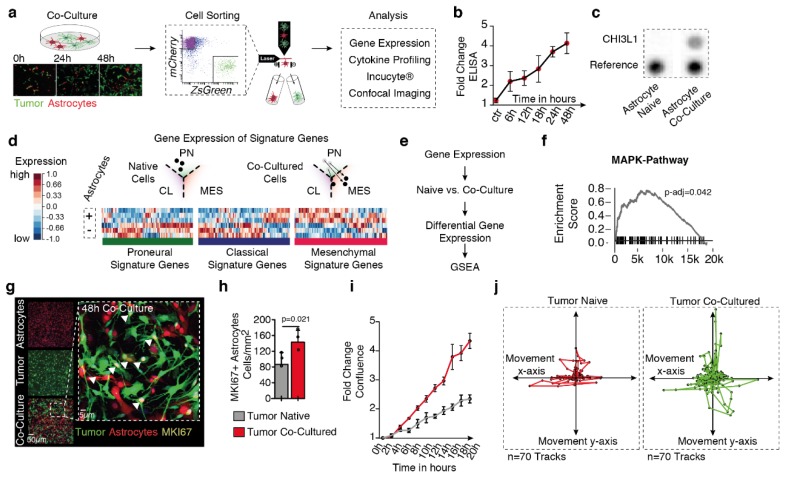
(**a**) Illustration of the workflow. Astrocytes and tumor cells (BTSC#168) are co-cultured and sorted by fluorescence-activated cell sorting (FACS). (**b**) ELISA assay reveals a time-dependent increase of CHI3L1 release in astrocytic-tumor co-culture. (**c**) Increase of CHI3L1 by proteome profile-assay. (**d**) Heatmap of gene-expression in glioblastoma subgroup signature genes. Up-regulated genes marked in red, down-regulated genes in blue. At the top, an illustration of subtype shift of co-cultured tumor cells towards the mesenchymal subgroup. (**e**,**f**) Gene set enrichment analysis of the MAPK-signaling pathway in co-cultured vs. naive cultured tumor cells. (**g**) Immunofluorescence imaging of the proliferation marker MKI-67 in naive and co-cultured tumor cells as well as quantification (**h**). (**i**) Time-dependent increase of cell confluence of tumor cells in naïve (grey) and co-cultured condition (red). (**j**) Illustration of cell movement measured by 70 single-cells tracks in naive (red) and co-cultured condition (green). *p*-values are determined by Wilcoxon sum rank test (**h**) adjusted by False Discovery Rate (FDR) (**f**). Data is given as mean ± standard deviation.

**Figure 3 cancers-11-01437-f003:**
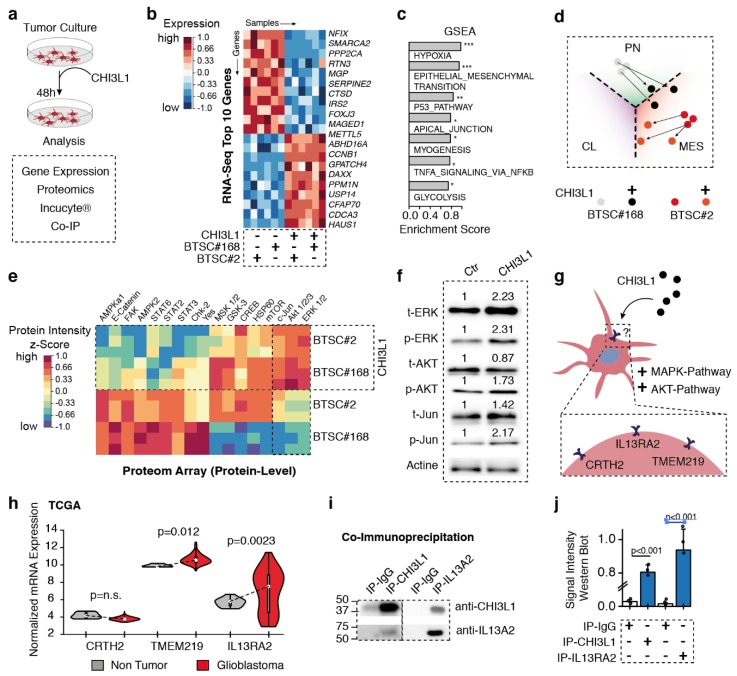
(**a**) Illustration of the workflow. (**b**) Heatmap of gene expression data after CHI3L1 treatment (top 10 up-, and down-regulated genes). Treatment was performed in a mesenchymal cell line (BTSC#2) and a proneural cell line (BTSC#168) (**c**) Gene set enrichment analysis after CHI3L1 treatment show increased signaling of hypoxia and epithelial–mesenchymal transition (EMT)-pathway related to the MAPK activation. (**d**) Illustration of the subtype shift of proneural and mesenchymal cell lines. (**e**) Heatmap of protein-level after CHI3L1 treatment show increased amount of c-Jun, Akt, and ERK phosphorylation, confirmed in western blot (**f**). (**g**) Illustration of potential binding partner of CHI3L1 based on literature reports. (**h**) Normalized expression of potential binding partners. IL13RA2 show the strongest difference between tumor and non-neoplastic samples (TCGA). (**i**) Co-immunoprecipitation of CHI3L1 and IL13RA2 reveals strong binding of CHI3L1-IL13Rα2. (**j**) Quantification of the co-immunoprecipitation. *P*-values are determined by one-way ANOVA (**h**,**j**) adjusted by adjusted by Benjamini–Hochberger (**h**,**j**). * *p* < 0.05, ** *p* < 0.01, *** *p* < 0.001, Data is given as mean ± standard deviation.

**Figure 4 cancers-11-01437-f004:**
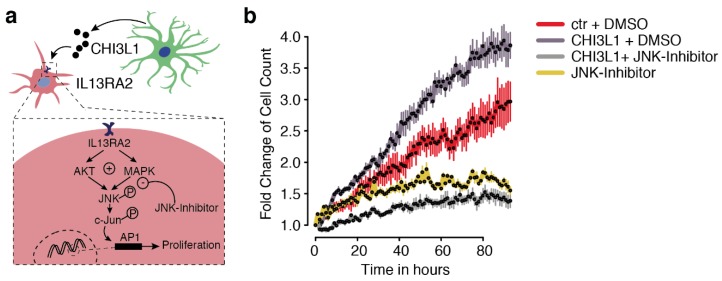
(**a**) Illustration of the potential CHI3L1 signaling in glioblastoma cells. (**b**) Cell proliferation measures by cell number within 80 h. Condition are given at the right side.

**Table 1 cancers-11-01437-t001:** List of primer.

Oligonucleotide Gene/Primer	Sequence	Source
TIMP1 forward	CTTCTGCAATTCCGACCTCGT	Thermo Fisher Scientific, Carlsbad, CA, USA
TIMP1 reverse	ACGCTGGTATAAGGTGGTCTG	Thermo Fisher Scientific, Carlsbad, CA, USA
VIM forward	AGTCCACTGAGTACCGGAGAC	Thermo Fisher Scientific, Carlsbad, CA, USA
VIM reverse	CATTTCACGCATCTGGCGTTC	Thermo Fisher Scientific, Carlsbad, CA, USA
STEAP4 forward	GAGAGTTCCGATTTGTCCAGTC	Thermo Fisher Scientific, Carlsbad, CA, USA
STEAP4 reverse	ATCTCTTCCCACCGTACACCA	Thermo Fisher Scientific, Carlsbad, CA, USA
CXCL10 forward	GTGGCATTCAAGGAGTACCTC	Thermo Fisher Scientific, Carlsbad, CA, USA
CXCL10 reverse	TGATGGCCTTCGATTCTGGATT	Thermo Fisher Scientific, Carlsbad, CA, USA
TGM1 forward	GCACCACACAGACGAGTATGA	Thermo Fisher Scientific, Carlsbad, CA, USA
TGM1 reverse	GGTGATGCGATCAGAGGATTC	Thermo Fisher Scientific, Carlsbad, CA, USA
SLC10A6 forward	GGAAGCTGTGGTCGCACAT	Thermo Fisher Scientific, Carlsbad, CA, USA
SLC10A6 reverse	GTAAAAGGCATGAGCCCAAACT	Thermo Fisher Scientific, Carlsbad, CA, USA
S100A10 forward	GGCTACTTAACAAAGGAGGACC	Thermo Fisher Scientific, Carlsbad, CA, USA
S100A10 reverse	GAGGCCCGCAATTAGGGAAA	Thermo Fisher Scientific, Carlsbad, CA, USA
EMP1 forward	GTGCTGGCTGTGCATTCTTG	Thermo Fisher Scientific, Carlsbad, CA, USA
EMP1 reverse	CCGTGGTGATACTGCGTTCC	Thermo Fisher Scientific, Carlsbad, CA, USA
CD14 forward	ACGCCAGAACCTTGTGAGC	Thermo Fisher Scientific, Carlsbad, CA, USA
CD14 reverse	GCATGGATCTCCACCTCTACTG	Thermo Fisher Scientific, Carlsbad, CA, USA
CD109 forward	AAGCCAGTGAAAGGAGACGTA	Thermo Fisher Scientific, Carlsbad, CA, USA
CD109 reverse	CCAGGGGAAGATAGATCCAGG	Thermo Fisher Scientific, Carlsbad, CA, USA
SRGN forward	AGGTTATCCTACGCGGAGAG	Thermo Fisher Scientific, Carlsbad, CA, USA
SRGN reverse	GTCTTTGGAAAAAGGTCAGTCCT	Thermo Fisher Scientific, Carlsbad, CA, USA
GBP2 forward	CATCCGAAAGTTCTTCCCCAA	Thermo Fisher Scientific, Carlsbad, CA, USA
GBP2 reverse	CTCTAGGTGAGCAAGGTACTTCT	Thermo Fisher Scientific, Carlsbad, CA, USA
AMIGO2 forward	CCTGGGAACCTTTTCAGACTG	Thermo Fisher Scientific, Carlsbad, CA, USA
AMIGO2 reverse	GCAAACGATACTGGAATCCACT	Thermo Fisher Scientific, Carlsbad, CA, USA
PSMB8 forward	CACGCTCGCCTTCAAGTTC	Thermo Fisher Scientific, Carlsbad, CA, USA
PSMB8 reverse	AGGCACTAATGTAGGACCCAG	Thermo Fisher Scientific, Carlsbad, CA, USA
FBLN5 forward	CTACTCGAACCCCTACTCGAC	Thermo Fisher Scientific, Carlsbad, CA, USA
FBLN5 reverse	TCGTGGGATAGTTTGGAGCTG	Thermo Fisher Scientific, Carlsbad, CA, USA
TAB1 forward	AACTGCTTCCTGTATGGGGTC	Thermo Fisher Scientific, Carlsbad, CA, USA
TAB1 reverse	AAGGCGTCGTCAATGGACTC	Thermo Fisher Scientific, Carlsbad, CA, USA
CD44 forward	CCACCCTAATCAAGGAAATGA	Thermo Fisher Scientific, Carlsbad, CA, USA
CD44 reverse	TGAAATCCAGGTGTTGGGATA	Thermo Fisher Scientific, Carlsbad, CA, USA
GFAP forward	CTG CGG CTC GAT CAA CTC A	Thermo Fisher Scientific, Carlsbad, CA, USA
GFAP reverse	TCCAGCGACTCAATCTTCCTC	Thermo Fisher Scientific, Carlsbad, CA, USA
CHI3L1 forward	CCA CCC TAA TCA AGG AAA TGA	Thermo Fisher Scientific, Carlsbad, CA, USA
CHI3L1 reverse	TGA AAT CCA GGT GTT GGG ATA	Thermo Fisher Scientific, Carlsbad, CA, USA
18S forward	TTT GCG AGT ACT CAA CAC CA	Thermo Fisher Scientific, Carlsbad, CA, USA
18S reverse	CCA CAC CCC TTA ATG GCA	Thermo Fisher Scientific, Carlsbad, CA, USA

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
