# Peer review of "Astrogliosis Releases Pro-Oncogenic Chitinase 3-Like 1 Causing MAPK Signaling in Glioblastoma"

_cancers, 2019, doi:10.3390/cancers11101437_

Round 1

Reviewer 1 Report

Wurm et al. performed gene expression analysis of purified astrocytes from tumor and non-infiltrated cortex revealed a tumor-related up-regulation of Chitinase 3-like 1 (CHI3L1), established and validated a co-culture model to investigate the impact of reactive astrocytes within the tumor microenvironment. They showed that reactive astrocytes promote a subtype-shift of glioblastoma towards the mesenchymal phenotype and drive MAPK signaling as well as proliferation and migration. They also demonstrated that MAPK signaling is directly caused by a CHI3L1-IL13RA2 co-binding, which leads to increased downstream MAPK and AKT signaling. This is a very interesting study, with minor comments: Figure 3f, it is better to show the signaling pathways in both cell models in western blotting. For Figure 3i, I would see the result by knocking down CHI3L1 or overexpression of CHI3L1 to confirm the binding.

Author Response

We thank the reviewers for their positive feed-back regarding our findings.

Rev1:

Wurm et al. performed gene expression analysis of purified astrocytes from tumor and non-infiltrated cortex revealed a tumor-related up-regulation of Chitinase 3-like 1 (CHI3L1), established and validated a co-culture model to investigate the impact of reactive astrocytes within the tumor microenvironment. They showed that reactive astrocytes promote a subtype-shift of glioblastoma towards the mesenchymal phenotype and drive MAPK signaling as well as proliferation and migration. They also demonstrated that MAPK signaling is directly caused by a CHI3L1-IL13RA2 co-binding, which leads to increased downstream MAPK and AKT signaling. This is a very interesting study, with minor comments:

Figure 3f, it is better to show the signaling pathways in both cell models in western blotting.

We thank the review for his supportive comments. Based on our findings from the protein array, we aimed to validate these findings by a conservative western blot approach. Due to the fact that BTSC#2 cells showed a high activation of the MAPK and AKT pathway at basline, we decided to use only BTSC#168 for further validation.

For Figure 3i, I would see the result by knocking down CHI3L1 or overexpression of CHI3L1 to confirm the binding.

Actually, we hypothesized that CHI3L1 is released by astrocytes. The co-binding was tested by recombinant CHI3L1 protein which was used to treat tumor cells. An additional CHI3L1 knockdown/silencing in astrocytes would be useful for the co-culture experiments. Unfortunately, we had problems to establish a stable gene silencing/overexpression in astrocytes, therefore we decided to use the recombinant protein for further experiments. Additionally, we aimed to investigate the specific effect of CHI3L1 on tumor cells, a gene silencing in astrocytes would lead to biased results through additional unknown effect of other cytokines released by astrocytes (like BDNF, Figure S3). These additional effects have to be explored in a future project.

Reviewer 2 Report

This report of how reactive astrocytes within the microenvironment of glioblastoma cells release a cytokine, CHI3L1, that drives MAPK signaling is very interesting.  Although it can be understood with careful reading what the authors are reporting, overall the usage of English is awkward which should be improved.  The figures illustrate interesting findings. However, Fig. S1's legend should explain the clusters more explicitly and use "clusters" when referring to more than one cluster. The legend for Fig. 3 should briefly describe the cell lines, BTSC#168 and BTSC#2, for the reader's convenience rather than only having the information in the text.  In the graph for Fig. 4b, the vehicle only for the JNK-inhibitor should be included if it was dissolved in a solvent such as ethanol or DMSO or anything other than water. Also, the cell viability in the presence of the JNK-inhibitor should be included to assure the reader that the results were specific for cell proliferation and not due to a toxic effect of the inhibitor.  Although this study focused on CHI3L1, the expression of BDNF is also interesting according to Fig. S3.  In Line 229, the authors probably meant to write "Figure S4" for the paper to make sense for me. The Materials & Methods section was easier to read than the rest of the paper in regard to the usage of English.  Some suggestions to consider for clarifying the content in regard to English include:  Line 36 "Investigation of the microenvironmental impact... Line 39 "...their regulation..."; Line 43 "...participates..."; Line 44 "...protects..."; Line 46 "...in maintaining...". I skipped the Results section for suggestions (too many).  More suggestions include Line 200 "...tumors..."; Line 201 "...glioblastomas..."; Line 202 "...was shown..."; Line 204 "...tumor cells and astrocytes..."; Line 209 "... aid by decreasing..."; Line 212 "...marked release of CHI3L1..."; Line 214 "...Alexander..."; Line 215 "...the increased release of CHI3L1 affects..."; Line 218 "...in vivo microenvironment..."  Line 220 "...have shown..."; Line 222 "...recently..."; Line 223 "...promotes..."; Line 224 "...the CHI3L1-related..."; Line 229 "...Therefore...". These are just a few suggestions to consider.

Author Response

Rev2:

This report of how reactive astrocytes within the microenvironment of glioblastoma cells release a cytokine, CHI3L1, that drives MAPK signaling is very interesting. Although it can be understood with careful reading what the authors are reporting, overall the usage of English is awkward which should be improved.

The figures illustrate interesting findings. However, Fig. S1's legend should explain the clusters more explicitly and use "clusters" when referring to more than one cluster.

We changed the requested explanations.

The legend for Fig. 3 should briefly describe the cell lines, BTSC#168 and BTSC#2, for the reader's convenience rather than only having the information in the text.

We add information regarding the cell lines in the figure description.

In the graph for Fig. 4b, the vehicle only for the JNK-inhibitor should be included if it was dissolved in a solvent such as ethanol or DMSO or anything other than water. Also, the cell viability in the presence of the JNK-inhibitor should be included to assure the reader that the results were specific for cell proliferation and not due to a toxic effect of the inhibitor.

We thank the reviewer for his comments. We used DMSO and add this information to the Figure 4. Additionally, we add a MTT assay to prove low toxicity for JNK inhibition. From the live imaging we showed that tumor cells still growing.

Although this study focused on CHI3L1, the expression of BDNF is also interesting according to Fig. S3.

Actually, there are more highly interesting released cytokines from astrocytes, here we aimed to focus on CHI3L1 but other cytokines will be investigated in further projects.

 In Line 229, the authors probably meant to write "Figure S4" for the paper to make sense for me.

This is correct, we changed the description

The Materials & Methods section was easier to read than the rest of the paper in regard to the usage of English.  Some suggestions to consider for clarifying the content in regard to English include:  Line 36 "Investigation of the microenvironmental impact... Line 39 "...their regulation..."; Line 43 "...participates..."; Line 44 "...protects..."; Line 46 "...in maintaining...". I skipped the Results section for suggestions (too many).  More suggestions include Line 200 "...tumors..."; Line 201 "...glioblastomas..."; Line 202 "...was shown..."; Line 204 "...tumor cells and astrocytes..."; Line 209 "... aid by decreasing..."; Line 212 "...marked release of CHI3L1..."; Line 214 "...Alexander..."; Line 215 "...the increased release of CHI3L1 affects..."; Line 218 "...in vivo microenvironment..."  Line 220 "...have shown..."; Line 222 "...recently..."; Line 223 "...promotes..."; Line 224 "...the CHI3L1-related..."; Line 229 "...Therefore...". These are just a few suggestions to consider.

We carefully went through the paper and improve the language.